# Subjective and objective effects of anxiety and fatigue on social function in patients with enterostomy and their family caregivers

**Ying Fan**[1], **Meixuan Song**[2], **Shicong Xu**[1‡], **Yuyue Tan**[1‡], **Xianrong Li**[2‡]*

**1** School of Nursing, Southwest Medical University, Luzhou, China, **2** Department of Gastrointestinal Surgery, Affiliated Hospital of Southwest Medical University, Luzhou, China

☯ These authors contributed equally to this work.
‡ SX, YT and XL also contributed equally to this work.
* 15775927757@qq.com

## Abstract

### Background

Many patients with enterostomy and their family caregivers experience severe anxiety and fatigue, which affects their participation in normal social activities and family life, resulting in impaired social function. The purpose of this study was to understand the status of social function of patients with enterostomy and their family caregivers;at the same time to analyze the subjective and objective effects of anxiety and fatigue on their social functions.

### Methods

The self-made general information questionnaire, WHO Social Disability rating Scale, Miao Yu's Multidimensional Fatigue Scale-20 and Zung et al.'s Self-rating Anxiety Scale were used to investigate the general situation, social function, fatigue and anxiety of enterostomy patients and their family caregivers who came to the hospital for treatment from March 28, 2023 to November 30, 2023. SPSS27.0 was used to complete the statistical analysis of the data, and AMOS26.0 was used to establish the structural equation model to complete the subject and object effect analysis.

### Results

A total of 260 pairs of enterostomy patients and their family caregivers were included in this study. The social function scores of enterostomy patients and their family caregivers were (8.80±3.44) and (6.44±3.60). The anxiety scores were (37.81±7.60) and (34.73±7.50). The fatigue scores were (63.35±12.80) and (51.21±12.38).The results of the subject-object effect analysis reported the subject effect: the anxiety of patients with enterostomy had a significant positive impact on the degree of social dysfunction ($\beta$ = 0.154, P = 0.015); The fatigue of patients with enterostomy had a significant positive effect on the degree of social dysfunction ($\beta$ = 0.132, P = 0.034). The anxiety of family caregivers had a significant positive effect on the degree of social dysfunction ($\beta$ = 0.161, P = 0.023). The fatigue of family caregivers had a significant positive effect on the degree of social dysfunction ($\beta$ = 0.201,

**Data Availability Statement:** All relevant data are available in the Open Science Framework repository at https://osf.io/muk7y/.

**Funding:** The author(s) received no specific funding for this work.

**Competing interests:** NO authors have competing interests.

P = 0.005). In terms of object effects: only the fatigue of family caregivers had a significant positive impact on the degree of social dysfunction of enterostomy patients (β = 0.224, P < 0.001), and other ways had no object effects.

## Conclusion

Patients with enterostomy and their family caregivers have serious social function defects. Clinical medical staff should take care of them as a whole in order to better return to family and society after surgery.

## Introduction

Colorectal Cancer, also known as large bowel cancer [1], refers to malignant tumors occurring in the colon and rectal epithelial cells, and is one of the most common cancers of the digestive system. According to the recent Statistics of Global Cancer Statistics 2020 [2], the incidence of colorectal cancer ranks the third in the world, second only to lung cancer and breast cancer. The mortality rate jumped to the second place, second only to lung cancer. According to the latest data from the National Cancer Center of China and the International Agency for Research on Cancer [3], the number of cases and deaths of colorectal cancer reached 571 thousand and 240 thousand in 2022, which has become the second most common cancer threatening men's health in China, second only to lung cancer. At present, surgical resection of tumor cells and their invaded intestinal and lymphatic vessels is the main treatment method for colorectal cancer [4], among which enterostomy is one of the important methods of surgical treatment. According to the Chinese colorectal cancer Surgery Case Registry Database, the 2020 annual report shows [5]: nearly 100,000 people in China undergo enterostomy surgery every year, and the number still has a trend of growth. Although enterostomy can delay the life of patients, once the patients have undergone the operation, they have to bear not only the pressure of cancer treatment, but also the psychological pressure of carrying enterostomy life and the pressure of stoma nursing. Therefore, they are prone to anxiety, depression, social avoidance and other physiological, psychological and social problems after surgery [6], which will have a serious impact on the social function of enterostomy patients and their family caregivers.

Patients after enterostomy are often troubled by negative psychological emotions due to diseases, among which anxiety is the most common [7, 8].Chen [9] investigated 98 patients with enterostomy after discharge, and the study showed that their incidence of anxiety was 41.83%.Liu [10] investigated the fatigue status of 52 pairs of enterostomy patients and their family caregivers, the results showed that the probability of fatigue in patients and their family caregivers within 6 months after surgery was 52.8% and 39.6%, respectively. And severe negative psychological emotions and fatigue will have a serious impact on the social function of enterostomy patients and their family caregivers. A study [11] investigated the social function of 126 patients with permanent colostomy, and found that the incidence of social function defects in patients with colostomy was as high as 80.16%.Therefore, it is necessary to explore the impact of anxiety and fatigue on social function of patients with enterostomy and their family caregivers. However, considering that patients and family caregivers are not completely independent from each other, they will be interrelated in daily interaction, so data analysis and result interpretation cannot be carried out through simple individual data. Therefore, in this study, the author used the Actor-Partner Interdependence Model [12] (APIM) to explore the relationship between social function, anxiety and fatigue.

APIM was proposed by American scholars Kenny and Cook [12] in 1999 (the standard model is shown in Fig 1), which is mainly used to analyze binary data and supplement the influencing factors of the dependent variable. The standard model of APIM covers four variables [13], as shown in Fig 1: where $Y_1$ and $Y_2$ are the dependent variables of the study individual in the pairwise relationship, respectively; $X_1$ and $X_2$ are predictor variables for the study individual in the pairwise relationship; $a_1$ and $a_2$ are the subject effect produced by the study individual in the pairwise relationship; $P_{21}$ and $P_{12}$ are the object effects that the study individuals have on each other in the pairwise relationship, and $E_1$ and $E_2$ represent the residual, which is the part of $Y_1$ and $Y_2$ that cannot be explained by $X_1$ and $X_2$. This model is considered to be the "gold standard" for analyzing pairwise data in pairwise relationships. Based on the theoretical and methodological basis, this study preliminarily explored the interaction between anxiety and fatigue on social function of enterostomy patients and their family caregivers, so as to explore more targeted dual level nursing measures.

## Materials and methods

### Design

This study was a cross-sectional study in which questionnaires were administered to patients undergoing enterostomy and their family caregivers. The study protocol was approved by the Ethics Committee of the Affiliated Hospital of Southwest Medical University (approval number: KY2023090);At the same time, informed consent was obtained from all the respondents before the investigation was carried out, and the letter was signed in person.

### Sample and setting

From March 28, 2023 to November 30,2023,convenience sampling was used to select patients with enterostomy and their family caregivers from Gastrointestinal surgery, anorectal Department, Oncology department and stoma clinic of two Class III Grade A hospitals in Luzhou, China as the research objects. Inclusion criteria of patients with enterostomy: (1) after the first enterostomy; (2) Colorectal cancer confirmed by pathological examination; (3) Correctly understand the survey content; (4) Informed consent for the study and voluntary

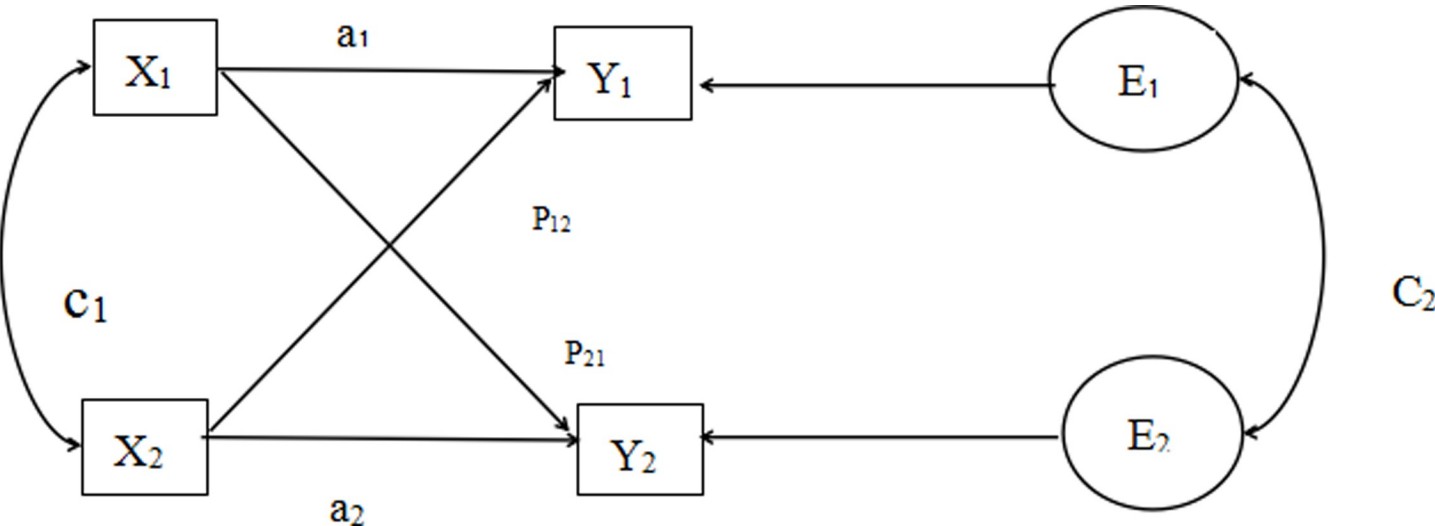

**Fig 1. Actor-partner interdependence model.**

participation; Exclusion criteria: (1) Patients with serious heart, lung, kidney, liver and other organ diseases; (2) accompanied by tumors in other parts; (3) People with cognitive impairment and mental illness. Inclusion criteria of family caregivers: (1) living with the patient and having a blood relationship with the patient; (2) be the main family caregiver of the patient without payment (care time ≥4 days a week); (3) Correctly understand the survey content; Exclusion criteria: (1) Employment relationship with patients; (2) complicated with serious diseases of heart, brain, kidney and other organs; (3) People with cognitive impairment and mental illness.

## Measurements

### (1) General information questionnaire

Social demographic data of enterostomy patients: age, gender, education level, marital status, occupation, living situation, family permanent residence, family per capita monthly income, number of children; Disease related information: after colostomy (months), medical payment method, clinical stage of cancer, whether cancer metastasis occurred, whether complicated with chronic diseases, whether there were complications of colostomy, whether radiotherapy and chemotherapy, and whether it was a permanent colostomy.

Social demographic information of family caregivers: age, gender, education level, marital status, occupation, living situation, family permanent residence, family per capita monthly income, number of children.

### (2) Social dysfunction screening scale(SDSS)

The scale is derived from the social function defect assessment tool developed by WHO in 1988 [14], which includes 10 items. Likert3 score is used to assign 0 to 2 points from "no defect" to "severe defect" respectively, with 0 points indicating normal social function, 1 point indicating partial defect of social function, and 2 point indicating serious defect of social function. A total score of < 2 was defined as normal social functioning; A score of ≥2 indicates a deficit in social functioning. The higher the total score, the more severe the patient's social function deficit. In this study, Cronbach's α coefficient was 0.72 for patients and 0.74 for family caregivers.

### (3) Self-rating Anxiety Scale (SAS)

This scale was developed by Zung [15] in 1971, and the frequency of anxiety related symptoms was evaluated by patients based on their own subjective symptoms. Cronbach's α of the scale was 0.913. There were 20 items in the scale, and each item was rated from "occasional/none" to "continuous" using the Likert4 rating method. In this study, Cronbach's α coefficient was 0.85 for patients and 0.85 for family caregivers.

### (4) The Multidimensional Fatigue Inventory-20(MFI-20)

Developed by Smets and Garssen in 1995 [16], it is a multidimensional scale to assess patient fatigue. There were 20 items in total, and each item was evaluated from "not at all" to "completely in line" by Likert5 scoring method, with a total of 100 points. The higher the score, the higher the degree of fatigue of the patient. In this study, Cronbach's α coefficient was 0.89 for patients and 0.92 for family caregivers.

## Statistical analysis

Statistical software SPSS27.0 and AMOS26.0 were used to analyze the data. The statistical analysis methods involved were as follows: descriptive analysis was used to describe the general

information distribution of the research objects, pearson correlation analysis was used to analyze the correlation between various variables. Paired t-test was used to compare the differences in social function, anxiety and fatigue scores between enterostomy patients and their family caregivers. AMOS26.0 was used to establish structural equation model to analyze the subject and object effects of anxiety and fatigue of enterostomy patients and their family caregivers on social function, and P < 0.05 was considered statistically significant.

## Results

### 1. General demographic and disease characteristics of enterostomy patients and their family caregivers

A total of 260 pairs of enterostomy patients and their family caregivers were included in this study. In this study, the age of patients with enterostomy was 28–94 years old, with an average age of (64.38±10.59) years old. The proportion of male patients (59.6%) was larger. Most of the patients (63.8%) had a primary school education or below, and a small number of patients (4.2%) had a college education or above. 80.3% of the patients were married. Most patients (46.2%) were retired, and most patients (92.7%) had permanent enterostomy.The age of family caregivers in this study ranged from 19 to 78 years old, with an average age of (52.05±12.51) years old. The number of female family caregivers (63.5%) was more dominant. Most of the family caregivers (41.9%) had a primary school education or below, and the number of family caregivers with college education or above (15.8%) was small. 81.9 percent of family caregivers were married. Most of the family caregivers (40.8%) are employed, and a small number (16.9%) are retired, See Table 1 for details.

### 2. Comparison of social function, anxiety and fatigue scores of enterostomy patients and their family caregivers

The social function scores of enterostomy patients and their family caregivers were (8.80 ±3.44) and (6.44±3.60). The scores of anxiety were (37.81±7.60) and (34.73±7.50). The fatigue scores were (63.35±12.80) and (51.21±12.38). The results of paired t-test showed that there were significant differences in the scores of social function, anxiety and fatigue between the two groups (P < 0.001), as shown in Table 2.

### 3. Correlation analysis of anxiety, fatigue and social function of enterostomy patients and their family caregivers

The results showed that the social function score of enterostomy patients was positively correlated with the social function score of family caregivers (r = 0.361,P < 0.01). The anxiety score of enterostomy patients was positively correlated with the anxiety score of family caregivers (r = 0.373,P < 0.01). The fatigue scores of enterostomy patients were positively correlated with those of family caregivers (r = 0.339,P < 0.01), as detailed in Table 3.

### 4. Subjective and object effects of anxiety, fatigue and social function in patients with enterostomy and their family caregivers

**4.1 Preliminary model construction and fitting results.** Based on the results of literature review and correlation analysis in this study, the subject and object interdependence model of this study was constructed in AMOS software with the anxiety and fatigue of enterostomy patients and their family caregivers as the independent variables and the social function of enterostomy patients and their family caregivers as the dependent variable. The initial model is shown in Fig 2.

**Table 1. General demographic data and disease data of enterostomy patients and family caregivers (n = 260).**

| Characteristics | Patients n(%) | Family caregivers n(%) |
|---|---|---|
| **Age** | | |
| ≤59 | 87(33.4) | 171(65.8) |
| ≥60 | 173(66.5) | 89(34.2) |
| **Gender** | | |
| Male | 155(59.6) | 95(36.5) |
| Female | 105(40.4) | 165(63.5) |
| **Education level** | | |
| Primary School and below | 166(63.8) | 109(41.9) |
| Junior high school | 61(23.5) | 49(18.8) |
| High school | 22(8.5) | 61(23.5) |
| College degree and above | 11(4.2) | 41(15.8) |
| **Marital status** | | |
| Married | 209(80.4) | 213(81.9) |
| Divorced/Widowed/Unmarried | 51(19.6) | 47(18.1) |
| **Career information** | | |
| On the job | 75(25) | 181(69.6) |
| Unemployment | 65(28.8) | 35(13.5) |
| Retirement | 120(46.2) | 44(16.9) |
| **Residence** | | |
| City | 134(51.5) | 134(51.5) |
| Rural area | 126(48.5) | 126(48.5) |
| **Per capita monthly household income (CNY)** | | |
| ≤1000 | 92(35.4) | 92(35.4) |
| 1001~ | 93(35.8) | 93(35.8) |
| 3001~ | 52(20.0) | 52(20.0) |
| >5000 | 23(8.8) | 23(8.8) |
| **Number of children** | | |
| ≤1 | 41(15.8) | 67(25.8) |
| 2 | 150(57.7) | 156(60) |
| ≥3 | 69(26.5) | 37(14.2) |
| **After enterostomy (month)** | | |
| 1–3 | 125(48.1) | |
| 3–6 | 99(38.1) | |
| >6 | 36(13.8) | |
| **Clinical stage of cancer** | | |
| I | 48(18.4) | |
| II | 118(45.4) | |
| III | 94(36.2) | |
| **Cancer metastasizes** | | |
| Yes | 148(56.9) | |
| No | 112(43.1) | |
| **Chronic disease** | | |
| Yes | 150(57.7) | |
| No | 110(42.3) | |
| **Complications of stoma** | | |
| Yes | 45(17.3) | |
| No | 215(82.7) | |

(*Continued*)

**Table 1.** (Continued)

| Characteristics | Patients n(%) | Family caregivers n(%) |
|---|---|---|
| **Radiotherapy and chemotherapy** | | |
| Yes | 135(51.9) | |
| No | 125(48.1) | |
| **Permanent stoma** | | |
| Yes | 241(92.7) | |
| No | 19(7.3) | |

In this study, relevant fitting indicators were used to evaluate the fitness of the model. The specific recommended values of the indicators and the fitting values of the initial model were shown in the following table. The results showed that the fitting index of the initial model in this study did not reach the recommended range, See Table 4.The path coefficients of the initial model are shown in Table 5.

**4.2 Model correction and fitting results.** According to the model modification suggestions and fitting results, the correlation between family caregiver anxiety and fatigue, family caregiver anxiety and fatigue of enterostomy patients, fatigue and anxiety of enterostomy patients, and family caregiver fatigue and anxiety of enterostomy patients were established. Moreover, three insignificant paths between family caregiver anxiety and patient social function, patient anxiety and family caregiver social function, and patient fatigue and family caregiver social function were deleted, and then the modified model was run, as shown in Fig 3. The fitting indicators of the modified model are all within the recommended range, as shown in Table 6.

The path coefficient results of the modified model show that in terms of the subject effect: The anxiety of patients with enterostomy had a positive effect on their social function ($\beta = 0.154$, P = 0.015), the fatigue of patients with enterostomy had a positive effect on their social function ($\beta = 0.132$, P = 0.034), and the anxiety of family caregivers had a positive effect on their social function ($\beta = 0.161$, P = 0.023). The fatigue of family caregivers had a positive effect on their social function ($\beta = 0.201$, P = 0.005). In terms of object effects, family caregiver fatigue had a positive impact on social function of enterostomy patients ($\beta = 0.224$, P < 0.001), and the results were shown in Table 7.

## Discussion

The results of this study showed that the score of SDSS in patients with enterostomy was (8.80 ±3.44),this result is similar to the result of Li [17] study on social function of patients with craniocerebral trauma. However, the score was higher than that of Cai [18] and Xiao [11] in the social function assessment of patients with urinary incontinence and patients with permanent enterostomy. The reasons for this phenomenon are as follows: firstly, patients with enterostomy have a strong sense of stigma after surgery due to changes in body image, odor and

**Table 2. Comparison of social function, anxiety and fatigue scores of enterostomy patients and their family caregivers (n = 260).**

| Characteristics | Patients($\bar{X}\pm S$) | Family caregivers($\bar{X}\pm S$) | t | P |
|---|---|---|---|---|
| SDSS | 8.80±3.44 | 6.44±3.60 | 9.548 | <0.001 |
| SAS | 37.81±7.60 | 34.73±7.50 | 5.864 | <0.001 |
| MFI-20 | 63.35±12.80 | 51.21±12.38 | 13.433 | <0.001 |

**Table 3. Correlation of social function, anxiety and fatigue among enterostomy patients and their family caregivers (n = 260).**

| variables | | patients | | | family caregivers | | |
|---|---|---|---|---|---|---|---|
| | | 1 | 2 | 3 | 1 | 2 | 3 |
| patients | 1 | 1 | | | | | |
| | 2 | 0.307** | 1 | | | | |
| | 3 | 0.290** | 0.419** | 1 | | | |
| family caregivers | 1 | 0.361** | 0.203** | 0.174** | 1 | | |
| | 2 | 0.236** | 0.373** | 0.165** | 0.288** | 1 | |
| | 3 | 0.321** | 0.362** | 0.339** | 0.297** | 0.597** | 1 |

**P<0.01,

1 = social function, 2 = anxiety, 3 = fatigue.

sound of colostomy excreta, which affects their social functions, A study [19] investigated the status quo and influencing factors of stigma of 75 patients with permanent enterostomy, and the results reported that the stigma of patients was at a moderate level, while severe stigma made patients have poor postoperative self-acceptance, resulting in strong negative emotions and affecting their normal social interaction [20]. In addition, patients with enterostomy feel unable to return to society normally after discharge, which affects their maintenance of interpersonal relationships with friends, relatives, and work partners, resulting in their inability to make full use of the help provided by others. Zhou [21] investigated the status of social participation of 110 patients with permanent enterostomy, and the results indicated that patients with enterostomy had low level of social participation, poor interpersonal communication skills, and low utilization of social support provided by the surrounding, which affected their normal communication with the outside world. Finally, it is more difficult for patients with enterostomy to transition from hospital to family or community due to stoma care, change of bowel habits and other reasons. The shorter the time after enterostomy, the higher the level of dependence on transitional care after discharge. Zheng [22] investigated the current status of transitional nursing dependence of 214 elderly patients with enterostomy, and compared the transitional nursing dependence level of patients at different time periods after surgery. The

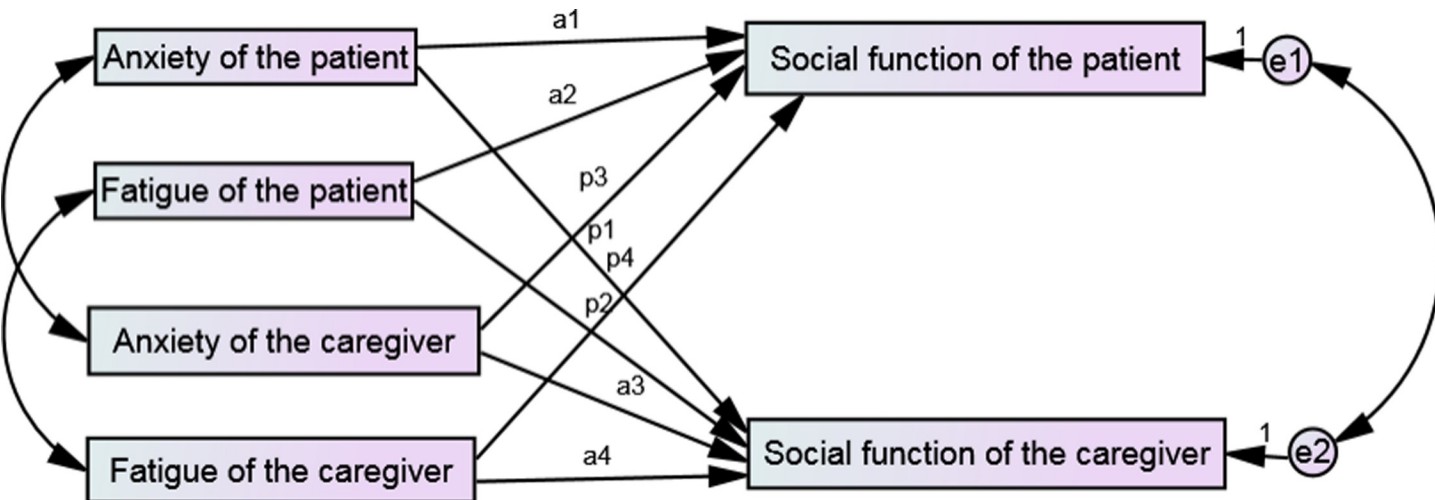

**Fig 2. Actor-partner interdependence model of social function, anxiety, and fatigue of enterostomy patients and their family caregivers.**

**Table 4. Initial model fitting index.**

| Indicators | Recommended value | Value of fit |
|---|---|---|
| $x^2$ | the smaller the better | 159.144 |
| $x^2/df$ | <3.0 | 39.798 |
| Goodness of Fit Index,GFI | >0.9 | 0.856 |
| Adjusted Goodness of Fit Index,AGFI | >0.8 | 0.242 |
| Root Mean Square Error of Approximation,RMSEA | <0.08 | 0.387 |
| Normed Fit Index,NFI | >0.9 | 0.512 |
| Incremental Fit Index,IFI | >0.9 | 0.518 |
| Comparative Fit Index,CFI | >0.9 | 0.501 |
| Tucker-lewis Index,TLI | >0.9 | 0.870 |

results indicated that the transitional nursing dependence level of patients within 3 months after enterostomy was significantly higher than that of patients over 3 months after surgery, and the difference was statistically significant. At the same time, a study [23] investigated and compared the quality of life of patients with enterostomy 3 months after surgery and 6 months after surgery. The results indicated that patients had a short discharge time, more psychological distress, a high level of psychological negativity, and a low level of stoma nursing mastery, which affected their overall social function. In this study, the majority of patients (48.1%) were in 1–3 months after enterostomy, which may be the reason for their higher score of social dysfunction than that of Xiao Bo et al. 's study.

The results of this study showed that the SDSS of family caregivers of enterostomy patients was (6.44±3.60).The score of this result is lower than that of Han [24] and Wang [25] in their assessment of social function of caregivers of hospitalized lung cancer patients and caregivers of patients with mental disorders. However, clinical workers should not ignore the attention paid to the social function of family caregivers of enterostomy patients. Some researchers [26] investigated the quality of life (including the social function dimension) of family members of permanent enterostomy patients, and the results showed that 90% of family members of enterostomy patients bear the burden of care after surgery, and their score of social function

**Table 5. Path analysis coefficients of the subject and object effects of the initial model.**

| | Standardized Estimate | S.E. | C.R. | P |
|---|---|---|---|---|
| P.social function<---P.anxiety | 0.164 | 0.030 | 2.390 | 0.017 |
| P.social function<---P.fatigue | 0.159 | 0.017 | 2.391 | 0.017 |
| P.social function<---C.anxiety | 0.039 | 0.033 | 0.522 | 0.602 |
| P.social function<---C.fatigue | 0.195 | 0.021 | 2.550 | 0.011 |
| C.social function<---P.anxiety | 0.057 | 0.032 | 0.815 | 0.145 |
| C.social function<---P.fatigue | 0.074 | 0.019 | 1.095 | 0.273 |
| C.social function<---C.anxiety | 0.166 | 0.036 | 2.159 | 0.031 |
| C.social function---C.fatigue | 0.160 | 0.022 | 2.408 | 0.041 |
| P.anxiety<--->C.anxiety | 0.373 | 3.768 | 5.620 | *** |
| P.fatigue<-->C.fatigue | 0.329 | 10.320 | 5.036 | *** |
| e1<-->e2 | 0.272 | 0.682 | 4.229 | *** |

P = patients; C = family caregivers;

*** P < 0.001;

S.E. = standard error; C.R. = critical ratio.

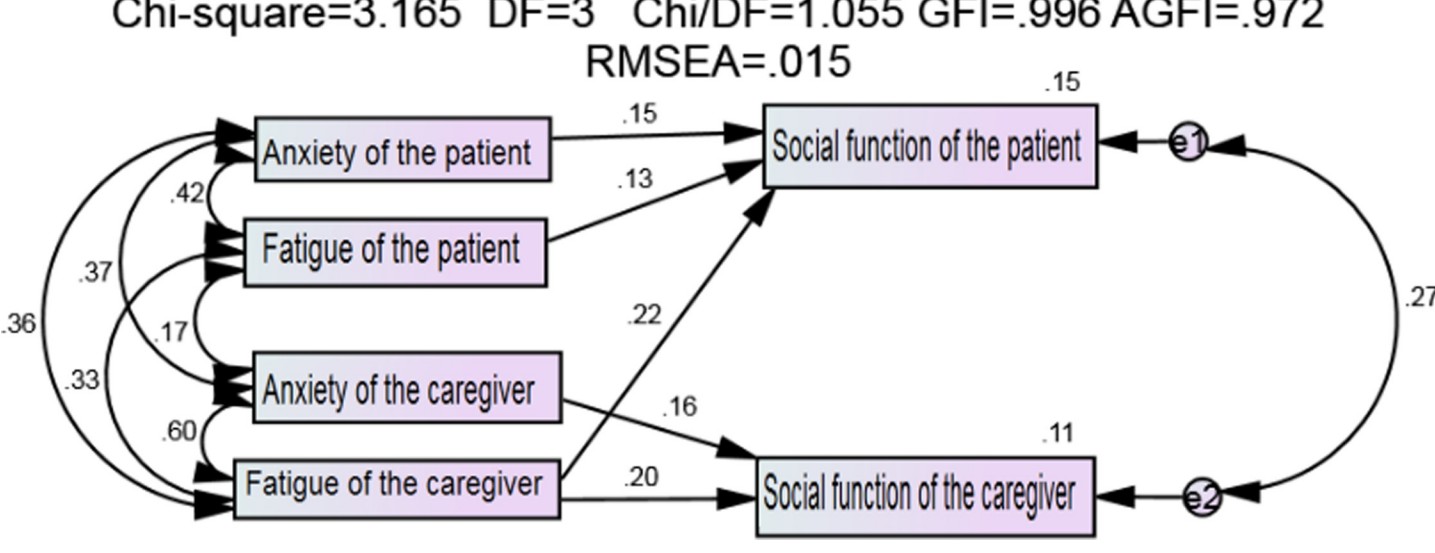

**Fig 3. Diagram of the modified Actor-partner interdependence model.**

dimension was the lowest in their quality of life. Rafiei [27] conducted an investigation and study on 250 family caregivers of enterostomy patients, and reported that they experienced serious impairment of their social function due to the interruption of their original lifestyle and financial problems. In view of this, it is suggested that we should not only pay attention to patients, but also pay attention to the social function of family caregivers in clinical work, actively guide family caregivers to express their concerns, and teach them relevant knowledge of ostomy care.

In this study, a structural equation model was developed to analyze the subject and object effects between anxiety, fatigue, and social functioning of enterostomy patients and their family caregivers. The results showed that the anxiety and fatigue of enterostomy patients and their family caregivers positively predicted the degree of social dysfunction. That is, the more serious the degree of anxiety, the more serious the social dysfunction; The more severe the degree of fatigue, the lower the level of social function, and vice versa, indicating that anxiety and fatigue are one of the influencing factors of social function. This result is consistent with the research results of Shi [28], Bao [29] and Luo [30], which all suggest that there is a positive correlation between anxiety and fatigue and the level of social dysfunction. The possible

**Table 6. The fit index of the modified model.**

| Indicators | Recommended value | Value of fit |
|---|---|---|
| $x^2$ | the smaller the better | 3.165 |
| $x^2/df$ | <3.0 | 1.055 |
| Goodness of Fit Index,GFI | >0.9 | 0.996 |
| Adjusted Goodness of Fit Index,AGFI | >0.8 | 0.972 |
| Root Mean Square Error of Approximation,RMSEA | <0.08 | 0.015 |
| Normed Fit Index,NFI | >0.9 | 0.990 |
| Incremental Fit Index,IFI | >0.9 | 0.999 |
| Comparative Fit Index,CFI | >0.9 | 0.999 |
| Tucker-lewis Index,TLI | >0.9 | 0.997 |

**Table 7. Path coefficients for the modified analysis of the subject-object effect.**

| | Standardized *Estimate* | S.E. | C.R. | P |
|---|---|---|---|---|
| P.social function<---P.anxiety | 0.154 | 0.028 | 2.282 | 0.015 |
| P.social function<---P.fatigue | 0.132 | 0.017 | 2.118 | 0.034 |
| P.social function<---C.fatigue | 0.224 | 0.017 | 3.582 | *** |
| C.social function<---C.anxiety | 0.161 | 0.034 | 2.282 | 0.023 |
| C.social function---C.fatigue | 0.201 | 0.021 | 2.782 | 0.005 |
| P.anxiety<-->C.anxiety | 0.373 | 3.768 | 5.620 | *** |
| P.fatigue<-->C.fatigue | 0.329 | 10.320 | 5.036 | *** |
| C.anxiety<-->C.fatigue | 0.597 | 6.692 | 8.248 | *** |
| C.anxiety<-->P.fatigue | 0.165 | 6.023 | 2.621 | 0.009 |
| P.anxiety<-->P.fatigue | 0.419 | 6.530 | 6.218 | *** |
| P.anxiety<-->C.fatigue | 0.362 | 6.194 | 5.480 | *** |
| e1<-->e2 | 0.274 | 0.691 | 4.225 | *** |

P = patients; C = family caregivers;

*** P < 0.001;

S.E. = standard error; C.R. = critical ratio.

reasons are as follows: after enterostomy, due to the change of patients' defecation habits, and the differences between the body structure and ordinary people, the patients have serious anxiety; For their family caregivers, they are worried about the patient's disease recovery, stoma healing and economic pressure after surgery, which makes them have great psychological pressure, and negative emotions of anxiety are full of their lives, further affecting their social function level. In addition, stoma care is a long-term and professional process for patients and their family caregivers. During this period, patients' laziness to their family caregivers increases, and family caregivers often bear more serious physical fatigue. It can also lead to severe mental fatigue, which further affects the level of social function of both. Negative emotions such as anxiety, severe fatigue and low level of social function will affect the physical health of patients and their family caregivers as well as the level of whole family function. This study analyzed only one object effect, that is, the social function of enterostomy patients was also affected by the fatigue of their family caregivers, however, the social function of family caregivers has not been found to be affected by the object factors of patients in this study. There may be other ways to influence their social functions, such as self-efficacy and mindfulness level [31]. Therefore, it is suggested that future researchers should take self-efficacy and mindfulness into account to continue to explore their influence on the social function of enterostomy patients and their family caregivers, and formulate intervention programs for their social function from a holistic perspective, so that they can better return to normal family and society after surgery.

## Conclusion

The social function of enterostomy patients and their family caregivers is seriously impaired, and their social function needs to be further improved. The social function of enterostomy patients is not only affected by their own anxiety and fatigue, but also affected by the fatigue of their family caregivers. The social function of family caregivers was only affected by their own anxiety and fatigue. The results of this model suggest that clinical nurses should provide nursing intervention for enterostomy patients and their family caregivers as a whole, so as to provide guidance and suggestions for improving their social function level.

## Supporting information

**S1 Raw data.**
(ZIP)

## Acknowledgments

Thank you to all the authors for your hard work.

## Author Contributions

**Conceptualization:** Xianrong Li.

**Data curation:** Ying Fan, Xianrong Li.

**Formal analysis:** Meixuan Song, Yuyue Tan.

**Funding acquisition:** Meixuan Song.

**Software:** Shicong Xu.

**Writing – original draft:** Ying Fan.

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
