## [Decision Letter · Decision Letter 0]

23 Aug 2024

Subjective and objective effects of anxiety and fatigue on social function in patients with enterostomy and their family caregivers

PONE-D-24-09812

Dear Dr. Li,

We’re pleased to inform you that your manuscript has been judged scientifically suitable for publication and will be formally accepted for publication once it meets all outstanding technical requirements.

Kind regards,

Boshra A Arnout

Academic Editor

PLOS ONE

Reviewers' comments:

Reviewer's Responses to Questions

**Comments to the Author**

1. Is the manuscript technically sound, and do the data support the conclusions?

Reviewer #1: Yes

Reviewer #2: Yes

2. Has the statistical analysis been performed appropriately and rigorously? 

Reviewer #1: Yes

Reviewer #2: Yes

3. Have the authors made all data underlying the findings in their manuscript fully available?

Reviewer #1: Yes

Reviewer #2: Yes

4. Is the manuscript presented in an intelligible fashion and written in standard English?

Reviewer #1: Yes

Reviewer #2: Yes

5. Review Comments to the Author

Reviewer #1: Dear Editor

Thank you for your invitation to review this manuscript

1-The research is good in terms of procedures

2-The research needs deeper recommendations

3-Sample selection was good

Therefore we can accept the manuscript as it is in the current form

Reviewer #2: The paper is methodologically sound, with appropriate tools and statistical techniques used to address the research questions. The results are clearly presented and discussed in the context of existing literature. The main limitation is the cross-sectional design, which restricts the ability to make causal inferences. Additionally, the convenience sampling method may limit the generalizability of the findings. Future research could benefit from a longitudinal approach and random sampling to address these limitations.

6. PLOS authors have the option to publish the peer review history of their article (what does this mean?). If published, this will include your full peer review and any attached files.

Reviewer #1: No

Reviewer #2: No

---

## [Editor Report · Acceptance letter]

6 Sep 2024

PONE-D-24-09812 

PLOS ONE

Dear Dr. Li, 

I'm pleased to inform you that your manuscript has been deemed suitable for publication in PLOS ONE. Congratulations! Your manuscript is now being handed over to our production team.

Kind regards, 

on behalf of

Professor Boshra A Arnout 

Academic Editor

PLOS ONE